# Preparation and Characterization of Transparent Polyimide–Silica Composite Films Using Polyimide with Carboxylic Acid Groups

**DOI:** 10.3390/polym11030489

**Published:** 2019-03-13

**Authors:** Kwan Ho Moon, Boknam Chae, Ki Seung Kim, Seung Woo Lee, Young Mee Jung

**Affiliations:** 1School of Chemical Engineering, Yeungnam University, Gyeongsan 38541, Korea; kwanho0903@gmail.com (K.H.M.); rltmdzz@naver.com (K.S.K.); 2Pohang Accelerator Laboratory, POSTECH, Pohang 37673, Korea; cbn@postech.ac.kr; 3Department of Chemistry, Kangwon National University, Chunchon 24341, Korea

**Keywords:** transparent polyimide–silica composite, sol–gel reaction, thermal imidization process, coefficient of thermal expansion, haze, yellowness index

## Abstract

Polyimide (PI) composite films with thicknesses of approximately 100 µm were prepared via a sol–gel reaction of 3-aminopropyltrimethoxysilane (APTMS) with poly(amic acid) (PAA) composite solutions using a thermal imidization process. PAA was synthesized by a conventional condensation reaction of two diamines, 3,5-diaminobenzoic acid (DABA), which has a carboxylic acid side group, and 2,2′-bis(trifluoromethyl)benzidine (TFMB), with 4,4′-(hexafluoroisopropylidene)diphthalic anhydride (6FDA) in N,N-dimethylacetamide (DMAc). The PAA–silica composite solutions were prepared by mixing PAA with carboxylic acid side groups and various amounts of APTMS in a sol–gel process in DMAc using hydrochloric acid as a catalyst. The obtained PI–silica composite films showed relatively good thermal stability, and the thermal stability increased with increasing APTMS content. The optical properties and in-plane coefficient of thermal expansion (CTE) values of the PI–silica composite films were investigated. The CTE of the PI–silica composite films changed from 52.0 to 42.1 ppm/°C as the initial content of APTMS varied. The haze values and yellowness indices of the composite films increased as a function of the APTMS content.

## 1. Introduction

To serve as flexible organic electronic device substrates, polymer materials should possess high gas barriers, flexibility, toughness, processability, thermal stability and chemical resistance [1,2,3]. Of the many available organic polymer materials, polyimides (PIs) have been widely used in the electronic device industry due to their excellent thermal properties, such as heat resistance, high thermal degradation temperature and high glass transition temperature, as well as their electrical and mechanical properties [4,5,6,7,8]. However, the optical properties of PI films have some disadvantages, such as low transmittance and a deep yellowish color, due to strong intermolecular interactions through pi–pi interactions and charge-transfer complex formation arising from the use of dianhydride and diamine monomers for polymerization [9,10]. Several concepts to create colorless PI films have recently been proposed that reduce charge-transfer complexation by introducing fluorine groups, asymmetrical and bulky pendent units, or alicyclic units into the polymer structure to overcome the poor optical properties of PI films [10,11,12,13,14,15,16]. However, the dimensional stability, glass transition temperature, thermal degradation temperature and coefficient of thermal expansion (CTE) of colorless PI films severely decrease in response to a reduction in intermolecular and intramolecular interactions in the polymer chains. To avoid the drawbacks of colorless PIs, hybridization of metalorganic compounds, which have organic ligands attached to metal atoms, such as Si, Al and Ti, with PI molecules has been introduced as a concept to obtain PIs with desirable thermal stability and chemical resistance [17,18,19,20,21,22].

Polymer composite films are defined as a combination of polymer matrices and either organic or inorganic fillers. Recently, polymer–inorganic composite materials fabricated via the sol–gel reaction method have received considerable attention due to the extraordinary thermal stabilities and mechanical properties of materials with small amounts of inorganic additives [23,24,25,26,27,28,29]. Among the various preparation methods for polymer–inorganic composites, sol–gel reactions have been widely examined. The sol–gel reaction method is advantageous because of its relatively low processing temperatures and the wide variety of metal-organic precursors that can be used [30,31,32]. Moreover, sol–gel reactions can produce nanostructured inorganic materials in polymer matrices and produce composite films with novel chemical and physical properties. Polymer–inorganic composite materials, which are nanoscale combinations of inorganic materials and polymers, are anticipated to be used in many applications, such as optics, membranes, protective coatings and electronics [10,25]. Therefore, polymer–inorganic composite films have been introduced as candidate substrate materials for the fabrication of next-generation electronic devices, including flexible displays, solar cells and printed circuit boards.

However, reports have shown that a high inorganic material content in the sol–gel process-induced phase separation for PI-composite films results in phase-separated materials with thermal stabilities and optical properties inferior to those of pure PI films [30,31,32,33,34]. Therefore, the preparation of PI-composite films with a small amount of inorganic material without reducing the thermal and optical properties of the resultant film is a major challenge for PI–inorganic hybridization via the sol–gel process. In previous works, we reported the thermal and optical properties of PI–silica composite films prepared using transparent poly(amic acid) (PAA) solutions, which were composed of a fluorinated dianhydride, 4,4′-(hexafluoroisopropylidene) diphthalic anhydride (6FDA); a fluorinated diamine, 2,2′-bis(trifluoromethyl)benzidine (TFMB); a carboxylic acid-functionalized diamine, 3,5-diaminobenzoic acid (DABA); and different amounts of (3-glycidyloxypropyl)trimethoxysilane (GTMS) and tetraethyl orthosilicate (TEOS) [35,36]. This paper reports the properties of transparent PI–silica composite films prepared using PAA with different amounts of 3-aminopropyltrimethoxysilane (APTMS) as the inorganic material. Here, the amine functional group in APTMS can form a covalent bond via a reaction with the carboxylic acid group in DABA. The goal of this study was to reveal the effect of APTMS in PI–silica composite films. The chemical structures of pure PI and PI-composite films were characterized using FTIR. Moreover, the optical and thermal properties of pure PI and PI–silica composite films were examined as a function of the APTMS content.

## 2. Materials and Methods

### 2.1. Synthesis of Poly(amic acid) 

Transparent PAA was synthesized by solution polymerization in a round-bottom flask filled with dry nitrogen gas [35,36]. In brief, 6FDA (10.00 g, 22.51 mmol) (Chriskev Company, Overland Park, KS, USA) was added into TFDB (5.77 g, 18.00 mmol) (Chriskev Company, Overland Park, KS, USA) and DABA (0.68 g, 4.50 mmol) (Merck KGaA, Darmstadt, Germany) and dissolved in dry DMAc (Merck KGaA, Darmstadt, Germany) by vigorous stirring to prepare the PAA (Figure 1). The reaction mixture was stirred continuously for 48 h. The solid contents of the synthesized PAA solutions were 15% (w/v). ^1^H NMR (DMSO-d6, δ): 10.8 (s, Ar-NH-), 10.3 (s, Ar-NH-), 8.4 (d, ArH), 8.2 (m, ArH), 8.0 (m, ArH), 7.9 (s, ArH), 7.8 (s, ArH). The inherent viscosity of the synthesized PAA was determined to be 0.67 dL/g at a concentration of 0.1 g/dL in DMAc at 25.0 °C. The obtained PAA solution was filtered through a syringe with a 1.0 µm filter and stored in a refrigerator.

### 2.2. Preparation of PI–Silica Composite Films. 

PAA–silica composite solutions were prepared via the sol–gel route by adding various quantities of APTMS with deionized water and a catalytic amount of HCl to the PAA solution [35,36]. The solid content of the prepared PAA–silica composite solutions was 10% (w/v). The APTMS contents applied in the PAA solutions were 2.5, 5.0 and 10.0 wt.% based on PAA. The PAA and APTMS solutions were stirred for 12 h at room temperature (Figure 1). A series of PI–silica composite films were prepared by further imidization of the PAA–silica composite solutions using a thermal imidization process. The PAA–silica composite solutions were coated onto Cr-coated steel plate substrates and dried at 80 °C for 1 h. The dried precursor films were imidized in a convection oven under a dry N_2_ gas flow using an imidization sequence (150 °C for 60 min, 200 °C for 60 min and 250 °C for 120 min) with a ramp rate of 2.0 °C/min. The prepared pure PI and PI–silica composite films were cooled to room temperature at a rate of 10 °C/min, which resulted in high-quality free-standing films with thicknesses of 97.9–105.7 µm.

### 2.3. Measurements. 

The inherent viscosity of the PAA was measured in DMAc at 25 °C using an Ubbelohde-type viscometer (Merck KGaA, Darmstadt, Germany). The FTIR spectra were recorded at a resolution of 4 cm^−1^ using a BOMEM DA8 spectrometer equipped with a liquid nitrogen-cooled MCT detector (BOMEM, Quebec, Canada) at the Pohang Accelerator Laboratory (PAL). The glass transition temperatures were measured from room temperature to 300 °C using a differential scanning calorimeter (DSC, model DSC-60C, Shimadzu, Japan). A ramp rate of 10.0 °C/min and dry N_2_ gas purging with a flow rate of 80 cc/min were used for the measurements. The thermal degradation temperatures were measured over a temperature range of 50–800 °C by a thermogravimetric analyzer (TGA, model TGA7, Perkin-Elmer, MA, USA) with dry N_2_ gas at a flow rate of 100 cc/min. The ramping rate for TGA measurements was 10.0 °C/min. The in-plane CTE of the samples was measured by a thermomechanical analyzer (TA Q400, TA Instruments, New Castle, DE, USA) with a 0.05 N expansion force applied to the film. For the measurement, a heating rate of 5 °C/min was applied in the temperature range of 50–250 °C. The CTE values were calculated in the temperature range (200–250 °C). The UV-visible spectra were recorded in transmittance mode using a Shimadzu UV-1800 spectrophotometer (Shimadzu Corporation, Kyoto, Japan). The haze values and yellowness indices of the composite films were determined using a Haze 4725 haze meter (Gardner Company, Reston, VA, USA) and a CM-3700d colorimeter (Minolta, Osaka, Japan), respectively.

## 3. Results and Discussion

The chemical structures of the pure PI film and PI–silica composite films were confirmed by FTIR spectroscopy (see Figure 2). The FTIR bands of the pure PI and PI–silica composite films were assigned in accordance with previously reported results [37,38,39,40,41,42,43,44]. Figure 2a shows the FTIR spectrum of the pure PI film. The bands corresponding to imide ring in PI main chain were located at 1786 (ν_s_(C=O)_imide ring_), 1724 (ν_as_(C=O)_imide ring_), and 1365 (ν_s_(CN)) cm^−1^, respectively. The characteristic bands of the CF_3_ group in the PI main chain were observed in the region of 1320–1180 cm^−1^ [33]. The characteristic bands from amide I (C=O stretching vibration) and amide II (NH bending vibration) in the PAA unit were not detected at approximately 1650 and 1550 cm^−1^, respectively, which indicated that the thermal imidization reaction was complete [33,35,41].

As shown in Figure 2, the bands corresponding to the PI main chain at 1786, 1724, and 1365 cm^−1^ were also observed for the PI–silica composite films. However, the characteristic bands assigned to the PAA unit were not observed. In contrast, the bands corresponding to the vibration modes, ν_s_(CH_2_) and ν_as_(CH_2_), at 2921 and 2850 cm^−1^ increased with increasing APTMS content. Further, the band at approximately 1100 cm^−1^ associated with the stretching vibration of the Si-O group also increased with increasing APTMS content in the series of composite films [32,33,42,43,44]. These results suggest that the PAA–silica precursors were completely converted to PI–silica composite films with the formation of silica networks in the PI matrix films. Furthermore, as the APTMS content increased, a new band at approximately 1700 cm^−1^ was observed as a shoulder. According to the report of Boroglu et al., the stretching vibration of the amide I group is observed at approximately 1680 cm^−1^ after the amidization of DABA and APTMS [42]. However, after imidization, it is not possible to resolve the band caused by amidization because it overlaps with the band attributed to the stretching vibration of the imide ring at approximately 1720 cm^−1^. Thus, the new band at approximately 1700 cm^−1^ observed in this study seems to be related to the reaction between DABA and APTMS.

Table 1 summarizes the compositions and appearances of the PI–silica composite films prepared by thermal imidization. The glass transition temperatures of the pure PI and PI–silica composite films were determined using DSC. A phase transition was not observed for both pure PI and PI–silica composite films over the temperature range of 25–300 °C, even after thermal treatment and repeated scanning. The thermal properties of the pure PI and PI–silica composite films were evaluated by TGA. The data are summarized in Table 1 and depicted in Figure 3. The pure PI and PI–silica-2.5 composite films exhibited two-step thermal degradation behavior. For the pure PI film, the first weight-loss step occurred at approximately 300 °C, which was associated with the loss of carboxylic acid groups from DABA in the main chain. The percentage of material lost in this first degradation step was consistent with the theoretical carboxylic acid group content based on the monomer feed composition [45,46]. The second weight-loss step occurred at approximately 450 °C and was ascribed to the decomposition of the aromatic component of the PI main chain. The PI–silica-2.5 composite film also exhibited two-step thermal degradation. The first weight-loss step occurred at temperatures as high as 250 °C and was related to the loss of a carboxylic acid group from DABA in the main chain; the second weight-loss step occurred at ~500 °C and was ascribed to the decomposition of the aromatic component of the PI main chain. Compared with the pure PI film, the PI–silica-2.5 composite film experienced the first weight-loss step at a lower temperature and the second weight-loss step at a higher temperature, which may be attributed to the remaining unreacted carboxylic groups and the formation of Si–O–Si, respectively. In contrast, the PI–silica-5.0 and PI–silica-10.0 composite films showed one-step thermal degradation that was dependent on the APTMS content. These results might be due to two factors: The disappearance of the carboxylic acid groups of DABA in the main chain because of new bond formation via a reaction with the amine groups in APTMS increased the decomposition temperature (*T*_d_) of the PI–silica composite films. The thermally stable silica networks which were produced by the sol–gel reaction with APTMS with a hydrochloric acid catalyst in the PI–silica composite matrix made improvement the thermal stability of the PI–silica composite films. Additionally, the residues of the composite films at 750 °C were 46.2–58.2 wt.% of the original weight, and the residues measured for the PI and composite films increased in the following order: PI < PI–silica-2.5 < PI–silica-5.0 < PI–silica-10.0.

The optical properties and transparency of the pure PI and PI–silica composite films were estimated by transmitted UV-visible spectroscopy using free-standing films with thicknesses of approximately 100 µm; the data are shown in Figure 4. Due to the thicknesses of the pure PI and PI–silica composite films, no transmittance was observed below 370 nm. Transparency was defined as the transmittance at a wavelength of 590.0 nm (Fraunhofer sodium D-line) for an unbiased comparison. Figure 4 shows that the PI and composite films were transparent obviously. The pure PI and PI–silica composite films revealed transparency with transmittance values over 85%. However, the transmittance values of the composite films slightly decreased with increasing APTMS content. The PI–silica composite films were prepared using APTMS as an inorganic sol–gel material to provide good transparency characteristics at an APTMS contents as high as 10 wt.%.

The CTE values of the series of composite films were measured by Thermomechanical Analyzer (TMA). The optical properties, including the yellowness indices and haze values, of the series of composite films were measured via colorimetry. Table 2 summarizes the yellowness indices, haze value and in-plane CTE values of the films. The in-plane CTE value of the pure PI film was 52.0 ppm/°C, and the CTE values of the PI–silica composite films were affected by the APTMS contents, decreasing with increasing APTMS content. The measured CTE values of the PI and PI–composite films decreased in the following order: pure PI > PI–silica-2.5 > PI–silica-5.0 > PI–silica-10.0. The decreasing trend of the CTE values might be due to APTMS network formation by the sol–gel reaction in the PI matrix. The silica networks formed in the PI-composite films reduce the mobility of the main chains in polymer. The CTE of the PI–silica-10.0 film was 42.1 ppm/°C, which was 19% lower than that of the pure PI film. The yellowness index and haze value of the pure PI film were 21.3 and 0.83%, respectively. The haze values of the PI–silica-2.5, PI–silica-5.0, and PI–silica-10.0 composite films were determined to be 1.14, 1.20 and 1.35%, respectively. The yellowness indices of the PI–silica composite films increased with increase of APTMS content. The measured yellowness indices of the pure PI and PI–silica composite films increased as follows: PI < PI–silica-2.5 < PI–silica-5.0 < PI–silica-10.0.

## 4. Conclusions

Polyimide (PI) composite films with thicknesses of approximately 100 µm were prepared via the sol–gel process of 3-aminopropyltrimethoxysilane (APTMS) with transparent poly(amic acid) (PAA) composite solutions using a thermal imidization process. Transparent PAA was synthesized by the conventional condensation reaction of two diamines, 3,5-diaminobenzoic acid (DABA) and 2,2′-bis(trifluoromethyl)benzidine (TFMB), with 4,4′-(hexafluoroisopropylidene)diphthalic anhydride (6FDA) in N,N-dimethylacetamide (DMAc). The PAA–silica composite solutions were prepared by mixing PAA possessing carboxylic acid side groups and various amounts of APTMS in a sol–gel process in DMAc using hydrochloric acid as a catalyst. PAA was synthesized with an inherent viscosity of 0.67 dL/g at a concentration of 0.1 g/dL in DMAc. PI–silica composite films obtained by the thermal imidization of the mixtures of PAA and different amounts of APTMS gave transparent free-standing films of good quality. The PI–silica composite films exhibited high thermal stability, and their degradation behaviors were found to be dependent on the APTMS content. The PI–silica composite films had a high decomposition temperature. The relationships among the APTMS content in the composite films, the optical properties and the in-plane CTE were examined. The haze values and yellowness indices of the films increased in the following order: PI < PI–silica-2.5 < PI–silica-5.0 < PI–silica-10.0. In contrast, the CTE decreased with the increasing APTMS content in the composite films, which suggests that the sol–gel reaction of APTMS contributes to the optical and thermomechanical properties of the PI–silica composite films. Optically transparent PI–silica composite films were obtained at APTMS contents as high as 10 wt.%. These findings propose that these PI composite films prepared using APTMS as an inorganic material are good candidate materials for flexible electronic substrate applications.

## Figures and Tables

**Figure 1 polymers-11-00489-f001:**
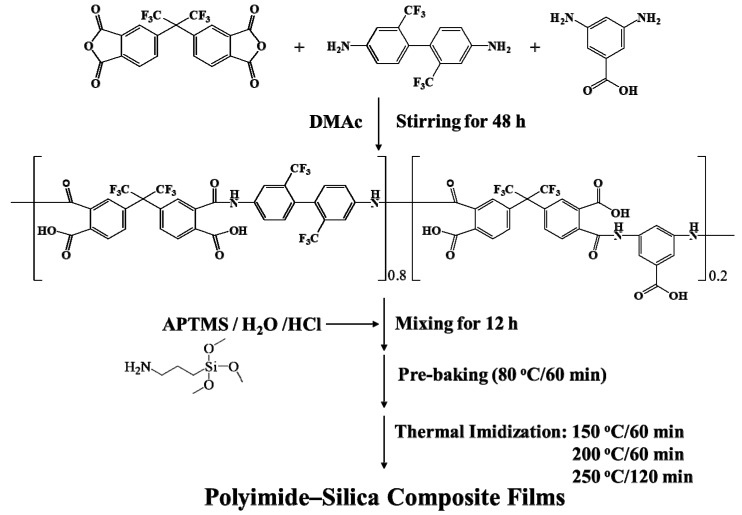
Chemical structure of poly(amic acid) (PAA) and the preparation procedure for the PI–silica composite films.

**Figure 2 polymers-11-00489-f002:**
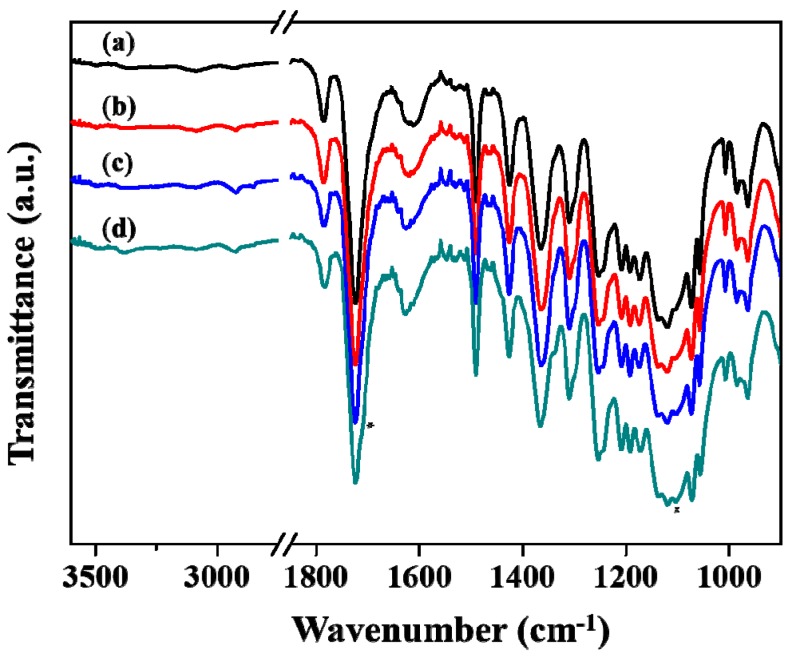
FTIR spectra of the PI and PI–silica composite films with various APTMS contents: (a) pure PI, (b) PI–silica-2.5, (c) PI–silica-5.0, and (d) PI–silica-10.0 films.

**Figure 3 polymers-11-00489-f003:**
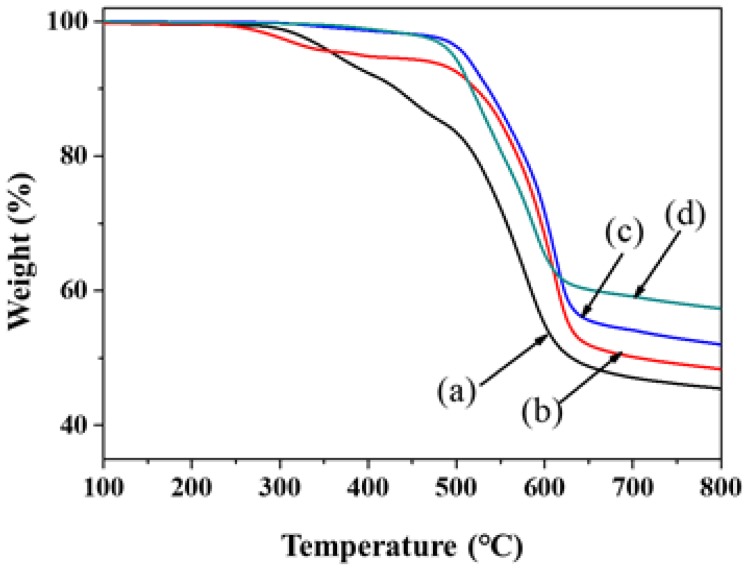
TGA thermograms of the pure PI and PI–silica composite films with various APTMS contents, which were measured at a heating rate of 10 °C/min: (a) pure PI, (b) PI–silica-2.5, (c) PI–silica-5.0, and (d) PI–silica-10.0 films.

**Figure 4 polymers-11-00489-f004:**
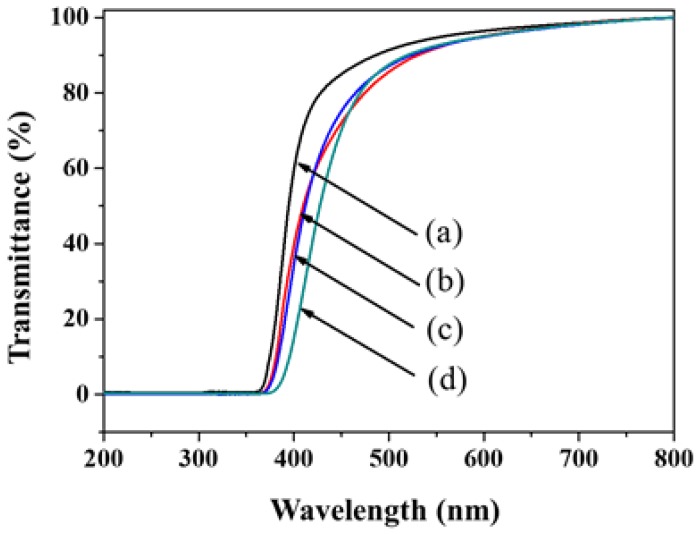
UV spectra of the pure PI and PI–silica composite films with various APTMS contents: (a) pure PI, (b) PI–silica-2.5, (c) PI–silica-5.0, and (d) PI–silica-10.0 films.

**Table 1 polymers-11-00489-t001:** Compositions, glass transition temperatures (*T*_g_), decomposition temperatures (*T*_d_), and residual weight percentages of the PI and PI–silica composite films.

Sample Designation	Feed Content of APTMS ^a^	*T*_g_ (°C) ^b^	*T*_d_ (°C) ^c^	RW (%) ^d^	Remarks ^e^
PI	-	- ^f^	364	46.2	T
PI–silica-2.5	2.5	-	393	49.2	T
PI–silica-5.0	5.0	-	510	53.0	T
PI–silica-10.0	10.0	-	497	58.2	T

^a^ Weight percent of APTMS based on the PAA feed during the preparation of the composite solutions. ^b^ Measured via DSC at a heating rate of 10.0 °C/min. ^c^ Determined via TGA at a heating rate of 10.0 °C/min under a nitrogen atmosphere; *T*_d_ = the temperature corresponding to a 5.0% weight loss. ^d^ Determined via TGA at a heating rate of 10.0 °C/min under a nitrogen atmosphere; RW = residual weight percentage at 750 °C. ^e^ T indicates that the film is transparent. ^f^ Not detected in the range 25–300 °C by DSC.

**Table 2 polymers-11-00489-t002:** Film thicknesses, yellowness indices, haze values, and CTE values of the PI and PI–silica composite films.

Sample Designation	Film Thickness (µm)	Yellowness Index ^a^	Haze (%)	CTE ^b^
PI	105.7	21.3	0.83	52.0
PI–silica-2.5	104.3	35.2	1.14	46.6
PI–silica-5.0	97.9	40.1	1.20	43.6
PI–silica-10.0	99.7	51.8	1.35	42.1

^a^ Measured in accordance with ASTM D 1925, E 308. ^b^ Measured via TMA at a heating rate of 5.0 °C/min. The CTE values were determined within the temperature range of 200–250 °C.

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
