# Peer review of "Preparation and Characterization of Transparent Polyimide–Silica Composite Films Using Polyimide with Carboxylic Acid Groups"

_polymers, 2019, doi:10.3390/polym11030489_

Round 1

Reviewer 1 Report

This manuscript by Lee and Jung et al. describes the synthesis and materials properties of polyimide-silica composites from a fluorinated dianhydride, a fluorinated diamine, a diamine that can coordinate to an amine-functionalized silica, and the silica precursor. This study seems very similar in nature to some of the previous papers from the same group, namely https://doi.org/10.1166/jno.2013.1525 and https://doi.org/10.1080/15421406.2013.844878, except the commercial silica source was swapped (one of them is cited in this paper, other not). Unfortunately I do not have access to these journals (still awaiting the library for copies), and would like to request the authors to provide a copy of these papers for reference. I believe the current paper is weak on the significance due to the lack of depth in discussion in light of previous research. The authors should critically review what the novelty of this work is: is it the use of 6FDA, the use of APTMS, or the combination of the two? In the end the results show that the addition of APTMS negatively affected the optical properties by increasing yellowness and haze, while decreasing CTE as expected and not changing the main aromatic chain thermal stability (the second step in the TGA), a somewhat expected result. To make up for this, a careful discussion outlining the intermolecular interactions and future directions to improve the materials properties need to be shown.

Some other comments to improve the manuscript are as follows:

- As the authors are probably aware, this field is quite rich in previous literature. I believe work from groups of J. E. Mark and G. L. Wilkes would be quite relevant and pioneering and thus should be cited.

- Can the degree of polymerization be characterized?

- The IR curves assignments need to be clearly stated. In particular, it would be interesting to observe the disappearance or weakening of free carboxylic acid groups by the addition of the aminosilanes: even if the amide C=O is overlapping, wouldn’t the free acid be visible at higher wavenumbers? Is that the one at 1780cm-1?

- It is unclear which weight composition (described in experimental) corresponds to which sample in Table 1. The feed ratio is in mol % or molar ratio? Ratio of what to what?

-The authors allude to previous literature (ref 28-30) for phase separation as a cause for reduction in optical, thermal and mechanical properties. In ref 29, as well as many of the work from Mark and Wilkes and others show distribution of silica in the polymer matrix using electron microscopy. Can you provide some electron microscopy to show the degree of homogeneity? This may allow the authors to give clues on optimizing the silica sol-gel conditions by tuning the sol particle sizes, for example.

Author Response

All the replies are written in bold.

We appreciate the reviewer’s comments on our manuscript, which have prompted us to significantly revise the manuscript.

This manuscript by Lee and Jung et al. describes the synthesis and materials properties of polyimide-silica composites from a fluorinated dianhydride, a fluorinated diamine, a diamine that can coordinate to an amine-functionalized silica, and the silica precursor. This study seems very similar in nature to some of the previous papers from the same group, namely https://doi.org/10.1166/jno.2013.1525 and https://doi.org/10.1080/15421406.2013.844878, except the commercial silica source was swapped (one of them is cited in this paper, other not). Unfortunately I do not have access to these journals (still awaiting the library for copies), and would like to request the authors to provide a copy of these papers for reference. I believe the current paper is weak on the significance due to the lack of depth in discussion in light of previous research. The authors should critically review what the novelty of this work is: is it the use of 6FDA, the use of APTMS, or the combination of the two? In the end the results show that the addition of APTMS negatively affected the optical properties by increasing yellowness and haze, while decreasing CTE as expected and not changing the main aromatic chain thermal stability (the second step in the TGA), a somewhat expected result. To make up for this, a careful discussion outlining the intermolecular interactions and future directions to improve the materials properties need to be shown.

To obtain colorless polyimide, a fluorinated diamine and/or dianhydride was applied for the synthesis. However, the polyimides obtained from the fluorinated monomer showed high CTE values and low thermal properties. Therefore, we tried to increase the thermal stability and decrease the CTE value. We previously attempted to prepare transparent polyimide composite films using (3-glycidyloxypropyl)trimethoxysilane (GTMS) and tetraethyl orthosilicate (TEOS) as inorganic materials for hybridization. In this work, we used 3-aminopropyltrimethoxysilane (APTMS) to prepare polyimide composite films. The purpose of this work was to examine the effect of the amine functional group of APTMS in a polyimide (after thermal imidization) with carboxylic acid groups. The carboxylic acid in the polyimide and the amine in the APTMS reacted to form a covalent bond, i.e., amide linkage, and trimethoxysilane provided sites for the sol-gel process. Comparing these three inorganic materials for hybridization, APTMS was the best inorganic material for preparing polyimide composite films with respect to the CTE, thermal stability and optical properties. Therefore, based on the reviewer’s comments, we added some sentences in the Introduction section in the revised manuscript.

Based on the Reviewer’s comment, the revised manuscript has been edited for proper English language, grammar, punctuation, spelling, and overall style by two highly qualified native English-speaking editors at American Journal Experts (www.aje.com).

The editorial certificate for the revised manuscript is attached as supporting information for the Editor.

Some other comments to improve the manuscript are as follows:

- As the authors are probably aware, this field is quite rich in previous literature. I believe work from groups of J. E. Mark and G. L. Wilkes would be quite relevant and pioneering and thus should be cited.

Based on the reviewer’s comments, we added four new references from the groups of J. E. Mark and G. L. Wilkes in the reference section as references 26 to 29 in the revised manuscript.

- Can the degree of polymerization be characterized?

Unfortunately, we could not determine the degree of polymerization. Poly(amic acid) molecules behave like polyelectrolytes in the eluent, resulting in poor GPC reproducibility. Therefore, the molecular weight of PAA can be determined by the inherent viscosity. The obtained PAA solution showed high viscosity during and after polymerization, and the inherent viscosity value (0.67 dL/g) indicated a relatively high molecular weight for PAA.

- The IR curves assignments need to be clearly stated. In particular, it would be interesting to observe the disappearance or weakening of free carboxylic acid groups by the addition of the aminosilanes: even if the amide C=O is overlapping, wouldn’t the free acid be visible at higher wavenumbers? Is that the one at 1780 cm-1?

Figure 2 shows the FTIR spectra of the polyimide film and polyimide composite films. It is not possible to observe the band originating from the carboxylic acid group in the polyamic acid precursor in this experiment.

In addition, it is not possible to observe the disappearance or weakening of the band associated with the free carboxylic acid groups in the benzoic acid unit upon the addition of aminosilane because of overlap with the imide C=O stretching at 1710 cm-1 (The characteristic bands originating from the aryl C=O stretching vibrations of a carboxylic acid group are observed at 1710–1680 cm−1). However, as described in the Results and Discussion section in this manuscript, it is possible to estimate the amidization of the DABA unit by addition and aminosilane. We described this in the following sentences in the manuscript.

Furthermore, as the APTMS content increased, a new band at approximately 1700 cm-1 was observed as a shoulder. According to the report of Boroglu et al., the stretching vibration of the amide I (C=O) group is observed at approximately 1680 cm-1 after the amidization of DABA and PTMS [42]. However, after imidization, it is not possible to resolve the band caused by amidization because it overlaps with the band attributed to the stretching vibration of the imide ring at approximately 1720 cm-1. Thus, the new band at approximately 1700 cm-1 observed in this study seems to be related to the reaction between DABA and PTMS.

- It is unclear which weight composition (described in experimental) corresponds to which sample in Table 1. The feed ratio is in mol % or molar ratio? Ratio of what to what?

The feed content of APTMS was based on the PAA content. Based on the reviewer’s comments, we corrected that in the text and Table 1 of the revised manuscript.

-The authors allude to previous literature (ref 28-30) for phase separation as a cause for reduction in optical, thermal and mechanical properties. In ref 29, as well as many of the work from Mark and Wilkes and others show distribution of silica in the polymer matrix using electron microscopy. Can you provide some electron microscopy to show the degree of homogeneity? This may allow the authors to give clues on optimizing the silica sol-gel conditions by tuning the sol particle sizes, for example.

Based on the reviewer’s comments, we tried several times to obtain an SEM image of the cross-section of the composite films. However, we did not obtain a high-quality image. Therefore, we took photographs of the pure PI and PI-composite films. As shown in the following photos, the yellowness increased with increasing APTMS content. The haze values increased with increasing APTMS content, but the films still showed good transparency.

Reviewer 2 Report

In the manuscript entitled “Preparation and Characterization of Transparent Polyimide-Silica Composite Films Using Polyimide Having Carboxylic Acid Groups” by Kwan Ho Moon et al., authors report optical and thermal properties of different polyimide composite films. They found that films prepared using sol-gel method are high-quality, transparent, possess high thermal stability including a high decomposition temperature. The papers present well described and discussed results.

I recommend the paper to publish in Polymers, after a certain revision due to the comments below.

My specific comments are listed below:

1.     I have difficulties to distinguish black lines in the figures (especially Figure 3 and 4). Please consider using color lines or different patterns.

2.     In the manuscript Table 1 is before Figure 3 which is confusing. In my opinion Table 1 summarizes information from Figure 2 and 3.  

Author Response

All the replies are written in boldface.

We appreciate the reviewer’s comments on our manuscript, which have prompted us to significantly revise the manuscript.

Comments and Suggestions for Authors

In the manuscript entitled “Preparation and Characterization of Transparent Polyimide-Silica Composite Films Using Polyimide Having Carboxylic Acid Groups” by Kwan Ho Moon et al., authors report optical and thermal properties of different polyimide composite films. They found that films prepared using sol-gel method are high-quality, transparent, possess high thermal stability including a high decomposition temperature. The papers present well described and discussed results.

I recommend the paper to publish in Polymers, after a certain revision due to the comments below.

My specific comments are listed below:

1.     I have difficulties to distinguish black lines in the figures (especially Figure 3 and 4). Please consider using color lines or different patterns.

According to the reviewer’s comment, we revised the Figures (Figure 2, 3 and 4) in the Results and Discussion section of the revised manuscript.

2.     In the manuscript Table 1 is before Figure 3 which is confusing. In my opinion Table 1 summarizes information from Figure 2 and 3.  

In this revised manuscript, we present the characteristics, including thermal properties, of a series of PI-silica composite films with varying aminosilane contents in Table 1 in advance. The feed ratio was calculated from the composition. Tg was calculated from the DSC curve (not shown in this manuscript). In addition, Td and RW were calculated from the TGA curve (Figure 3). Then, we described the TGA curve (Figure 3) to estimate the different characteristics of a series of PI-silica composite films depending on the aminosilane content in detail. Thus, we use Table 1 to show the representative values depending on the aminosilane content before Figure 3.

In addition, based on the Reviewer’s comment, the revised manuscript has been edited for proper English language, grammar, punctuation, spelling, and overall style by two highly qualified native English-speaking editors at American Journal Experts (www.aje.com).

The editorial certificate for the revised manuscript is attached as supporting information for the Editor.

Round 2

Reviewer 1 Report

I truly appreciate the extensive effort the authors put into revising the manuscript, and recommend publication of this manuscript in the current form.